# Theoretical and Experimental Study of Different Side Chains on 3,4-Ethylenedioxythiophene and Diketopyrrolopyrrole-Derived Polymers: Towards Organic Transistors

**DOI:** 10.3390/ijms25021099

**Published:** 2024-01-16

**Authors:** Shiwei Ren, Wenqing Zhang, Jinyang Chen, Abderrahim Yassar

**Affiliations:** 1Advanced Materials Laboratory, Zhuhai-Fudan Innovation Institute, Guangdong-Macao in-Depth Cooperation Zone in Hengqin, Hengqin 519000, China; shiwei_ren@fudan.edu.cn; 2Key Laboratory of Organic Solids, Institute of Chemistry, Chinese Academy of Sciences, Beijing 100190, China; zhangwq@iccas.ac.cn; 3Laboratory of Physics of Interfaces and Thin Films, Institut Polytechnique de Paris, 91128 Palaiseau, France

**Keywords:** copolymers, Stille coupling polycondensation, coplanarity, conjugated and non-conjugated, hole mobility

## Abstract

In this research, two polymers of P1 and P2 based on monomers consisting of thiophene, 3,4-Ethylenedioxythiophene (EDOT) and diketopyrrolopyrrole (DPP) are designed and obtained via Stille coupling polycondensation. The material shows excellent coplanarity and structural regularity due to the fine planarity of DPP itself and the weak non-covalent bonding interactions existing between the three units. Two different lengths of non-conjugated side chains are introduced and this has an effect on the intermolecular chain stacking, causing the film absorption to display different characteristic properties. On the other hand, the difference in the side chains does not have a significant effect on the thermal stability and the energy levels of the frontier orbitals of the materials, which is related to the fact that the materials both feature extremely high conjugation lengths and specific molecular compositions. Microscopic investigations targeting the side chains provide a contribution to the further design of organic semiconductor materials that meet device requirements. Tests based on organic transistors show a slight difference in conductivity between the two polymers, with P2 having better hole mobility than P1. This study highlights the importance of the impact of side chains on device performance, especially in the field of organic electronics.

## 1. Introduction

In recent years, there have been numerous studies on the use of organic materials as new semiconductor materials in the field of organic electronics, which is directly related to the chemically modifiability of organic materials [1,2,3,4]. Functionalized device applications in the field of organic electronics generally include organic light-emitting diodes, organic solar cells, organic electrochemical transistors, organic thermoelectrics, organic sensors, organic field-effect transistors (OFETs), and so on [5,6,7,8]. In order to realize functionalized applications and continuously improve device performance, it is necessary to design and prepare novel materials with suitable molecular structures [9,10,11,12,13]. A variety of organic molecular structures with the advantages of a low weight, flexibility, low cost, and solution processability have been developed, especially for organic polymer materials [14,15,16]. One class of widely studied systems is based on polythiophene, which has good electron-donating abilities and serves as one of the most essential materials for hole transport. In the case of poly(3-hexylthiophene) (P3HT), for example, the polymerization process is affected by the catalyst and other experimental conditions. It is often criticized for its tendency to form a variety of isomers such as head–tail and head–head isomers, which affects the regularity of the material [17,18]. The coplanarity of the molecular structure in the material facilitates the formation of good stacking and promotes the intra- and inter-chain transport of carriers to improve the performance of the device [19,20]. To improve the overall planarity of the backbone, various methods based on chemical molecular structure modification have been developed. First, conformational control groups such as alkenes and alkynes can be inserted between thiophene and thiophene to limit the torsion of thiophene. A typical example is the thiophene–vinylene–thiophene (TVT) structure [21,22]. Liu et al. reported a polymer of the DPP-TVT class with very close π − π stacking distances and good electron mobility [23]. Thiophene can then also be fused with other aromatic rings to form heteroaromatic groups, such as benzothiophene and selenothiophene, in order to improve the coplanarity of the backbone [24]. Biewer et al. recently reported polymers based on the composition of DPP and thieno [3,2-b]pyrrole, and showed 10^−2^ cm^2^ V^−1^ s^−1^ of moderate hole mobility. Finally, the formation of intramolecular conformational locks through weak intramolecular forces is also a method of interest. Intramolecular hydrogen bonding, S…O forces, S…F forces, electrostatic potential gravity, van der Waals forces, etc., are potential building systems [25,26,27]. Weak interaction forces tend to be flexible compared to TVT structures or heteroaromatic structures with a rigid architecture. Weak interaction forces and weak crystallinity are often required for self-recovering and self-repairing functions in materials and devices that are oriented towards stretchability or foldability, thus achieving a balance between electrical and mechanical properties [28]. Wang et al. prepare stretchable transistor devices by physically mixing semiconductor-conjugated polymer P3HT-like components with elastomeric components [29]. Hsu et al. provide ideas for soft electronic devices through a soft-end–hard-end combination strategy [30].

In this study, we have further expanded the electron-donating electrons of thiophene monomers by introducing an electron-rich group of 3,4-Ethylenedioxythiophene (EDOT) [31,32,33,34]. At the same time, we expect to use the introduced O atoms to form more intramolecular weak forces to control the conformation and planarity of the materials (Figure 1). We used the structure of EDOT with a common acceptor building block diketopyrrolopyrrole (DPP) to prepare polymers P1 and P2 using the Stille polymerization method [35,36,37]. Studies in recent years have also demonstrated the direct effect of side chains on material properties and device performance [38,39]. The objective of this study was to investigate the effect of side chains on the physical properties and electrical conductivity of such polymers, based on the use of two DPP units with different alkyl chains (R).

## 2. Results

### 2.1. Synthesis Routes to Polymers P1 and P2 through Stille Coupling Polymerization

The synthetic route for the preparation of the polymer via a palladium (Pd)-catalyzed Stille coupling reaction of the two monomers is shown on the left of Figure 1. The tris (dibenzylideneacetone) dipalladium, often abbreviated to Pd_2_(dba)_3_, is used as the catalyst. In order to stimulate the catalyst’s activities, the addition of a phosphorus ligand is required to reduce the divalent palladium to a zero-valent one. Subsequently, the C–Br bond in the monomeric DPP-Br-C8C10 or DPP-Br-C10C10 breaks readily and undergoes an oxidative addition reaction with the Pd coordinates, which initiates the reaction and starts the cycle (right side of Figure 1). The choice of chlorobenzene solvent was related to the solubility of the polymer. While the DPP monomer showed good solubility in conventional solvents due to its long alkyl side chains, the polymer was only soluble in chlorinated solvents due to its extremely high molecular weight. The same chemical equivalents and reaction time controls were used to analyze the differences in the final products of the reaction. The polymerization reaction was terminated after 12 h and purified via Soxhlet extraction [40]. The low-molecular-weight oligomers were extracted sequentially through methanol, acetone and hexane. The target materials were finally collected in chloroform in yields of 88% and 79%, respectively. Polymers P1 and P2 showed high solubility in halogenated solvents such as chloroform or chlorobenzene, which have a dark green color and limited solubility in common organic solvents such as ethyl acetate. The solubility in both chloroform and chlorobenzene exceeded 5 mg/mL, so these solvents were used for subsequent processing. The molecular weights of the materials were characterized using high-temperature gel permeation chromatography and the results are summarized in Table 1. The dispersive indexes (Đ) of the two polymers were comparable, but their number average molecular weights (Mn) and weight average molecular weights (Mw) varied considerably. The viscosity average molecular weight (Mv), the Z average molecular weight (Mz), the peak average molecular weight (Mp) and the Z + 1 average molecular weight (Mz_+1_) of the polymers are listed in Appendix A and are shown in the usual order Mn < Mv < Mw < Mz < Mz_+1_. The molecular weight of polymer P1 was found to be approximately twice that of P2 (Mw of 54.4 k and 27.4 k, respectively). Based on the ratio of Mw to the molecular weight of the smallest repeating unit in the polymer chain, the average chain lengths of the two polymers are assumed to be 54 and 24, respectively. An elemental analysis of the polymer solids verified the composition and purity of the materials, the results of which are shown in Table 1 below.

In order to compare the thermodynamic stability of the two polymers, thermogravimetric analyses (TGA) and differential scanning calorimetry (DSC) were carried out and the curves are shown in Figure 2a,b and Appendix A, respectively. Although the polymers have different chain lengths, they both exhibit similar stability in the range of 30–370 °C, with very little decomposition. Specifically, P1 loses 5% of its weight at temperatures approaching 380 °C, while P2 loses 5% of its weight at 385 °C. We believe that this small difference may be related to the length of the side chains of the material. The thermal stability of conjugated materials beyond a certain length of conjugation is mainly related to the molecular composition of the side chains, but is not proportional to the length of conjugation [41]. The results of the infrared (IR) patterns are shown in Figure 2c,d below, where the IR absorptions of the materials with different conjugation lengths are similar. Their typical carbon–oxygen (C–O) bonding and carbonyl characteristic (C=O) peaks appear near 1080 cm^−1^ and 1655 cm^−1^, respectively. The characteristic peaks of long aliphatic chains are obvious and found in the range of 2850–2920 cm^−1^.

### 2.2. Photochemical Properties

The ultraviolet–visible absorption of P1 and P2 was carried out in dilute solution and in the solid state of the thin film, respectively, and the results are shown in Figure 3 below. Their absorption phenomena in solution are almost the same, and both exhibit two absorption bands, which is due to the same molecular composition of the conjugated structure of the two materials. The relatively low absorption band in the 350–500 nm range originates from the π-π* transition, while the significantly higher absorption intensity in the 700–1100 nm range is due to the intramolecular charge transfer from the donor to the acceptor. P1 displays a maximum absorption (λ_max_) peak at 952 nm and a shoulder peak near 874 nm in solution. The λ_max_ peak of P2 in solution does not deviate from the maximum absorption peak of P1 by more than 5 nm (Table 2). On the other hand, the absorption of the two materials in the film phase is quite different. P2 still exhibits a maximum absorption peak as well as a pronounced shoulder peak, whereas P1 does not feature the tiny shoulder. The onset of absorption (λ_onset_) in the solid state produces a redshift of approximately 50 nm with respect to the λ_onset_ in the solution state, which leads to a significant decrease in the energy level. The energy bandgap of P2 in the solid-state film is only 1.12 eV, which indicates a better π-π stacking in the film state.

### 2.3. Electrochemical Properties

In order to further investigate the effect of varying the side chains of the materials on their redox properties and the energy levels of the frontier molecular orbitals (FMO), electrochemical tests based on cyclic voltammetry were conducted, as shown in Figure 4 below. Both polymeric materials show quasi-reversible oxidation characteristics, with peaks of 0.99 V and 1.01 V. Simultaneously, they exhibit similar reduction profiles and possess peaks of −0.93 V and −0.94, respectively. Depending on the onset of the oxidation peak and the onset of the reduction peak (0.77 V and −0.66 V, respectively), we can estimate that the corresponding highest occupied molecular orbital (HOMO) and lowest unoccupied molecular orbital (LUMO) energy levels of P1 are −5.17 eV and −3.74 eV, respectively. The energy levels of the HOMO and LUMO of P2 are extremely close to those of P1, at −5.20 eV and −3.76 eV, respectively. Our results from this test show that, despite the slight differences in the side chains, they do not have a drastic effect on the FMO energy levels of the materials. Aliphatic side chains consisting of non-conjugated components do not contribute significantly to the energy level of the conjugated main chain fraction [39,42].

### 2.4. Density Functional Theory (DFT) Calculations

To further characterize the intramolecular conformation of the polymers, DFT calculations were used based on Gaussian 16 software [43]. For the geometry optimization and frequency calculations, the B3LYP functional and the def2-SVP basis set was used, and the optimal geometry for each compound was determined [44,45]. The additional DFT-D3 London dispersion was introduced and analyzed in order to obtain accurate details of the intramolecular structure [46]. We used isopropyl chain-substituted dimers as computational models in order to balance the computational cost with the need to maximize the microscopic contribution of side chains to the structure [47]. The representative intramolecular atomic distances and dihedral angles between fragments are presented in Figure 5a and Figure 5b, respectively. The DPP unit consists of two five-member fused aromatic rings and exhibits good coplanarity. The DPP ends are linked to the thiophene by a single bond and exhibit a small dihedral angle of about 16°. The distance between the oxygen atom (O) in the carbonyl group and the hydrogen atom (H) in the neighboring thiophene is only 2.1 Å, which is smaller than the van der Waals force radius and thus facilitates the formation of intramolecular hydrogen bonds. The sulfur atom (S) on thiophene shares a S…O distance of 2.9 Å with EDOT, which facilitates the formation of an intramolecular conformational lock and thus further stabilizes the conformation of the material. The weak non-covalent bond interactions analysis shown in Figure 5c further validates the attraction of the stable configuration present within the molecule. Intramolecular reduced density gradient (RDG) analyses revealed a pronounced bluish-green coloration across their corresponding positions, demonstrating these attractive interactions [48]. The oxygen of the carbonyl group forms an intramolecular hydrogen bond with the hydrogen atom on thiophene, which appears blue-green in color. The S atoms have a pronounced van der Waals gravitational attraction with the H atoms on the alkyl chain. The thiophene unit interacts with the EDOT unit via the S…O interactions and S…H interactions to form a flat coplanar structure. The electrostatic surface potential (ESP) of the material is characterized in Figure 5d, where the carbonyl position is surrounded by a clear attraction of electrons; meanwhile, the EDOT ring shows a donation feature. The results of the orbital energy level analysis of the dimer are depicted in Figure 5e below, and its corresponding HOMO and LUMO energy levels are calculated to be −5.10 eV and −3.51 eV, respectively. It is worth mentioning that the HOMO energy levels calculated using theoretical simulations are close to those derived from the electrochemical tests. The theoretical and experimental values of the LUMO energy levels are slightly different, resulting in a difference of about 0.15 V between the bandgaps obtained by the two techniques. The orbital diagrams and the corresponding HOMO − 1/LUMO + 1 and HOMO − 2/LUMO + 2 are shown in Appendix A. Appendix A demonstrates the theoretically simulated absorption characteristic curve of the dimer, accompanied by the maximum absorption peak that appears near 815 nm. The onset absorption peaks are in close agreement with the actual absorption measurements, both appearing in the vicinity of 1050 nm. Appendix A lists the spatial coordinates of all the atoms of the dimer.

### 2.5. OFET Device Performance

In order to determine the difference in conductivity between the two materials in terms of charge transfer, we fabricated bottom gate bottom contact (BGBC) structured OFET devices based on polymers P1 and P2. The structure of the devices is shown in Appendix A, and the preparation protocols of the OFETs are placed in the Materials and Methods section. The devices based on P1 or P2 demonstrate hole mobility but no electron mobility, which is typical of unipolar P-type organic semiconductor materials. Table 3 shows the hole mobility extracted from the transfer characteristic curve (Figure 6a–b), and it can be seen that P2 outperforms P1 in terms of device performance. The devices using P2 have a saturation hole mobility that is about ten times higher than that of P1 and benefit from a high on/off current ratio (I_on_/I_off_).

### 2.6. Microstructure and Morphological Analysis of Polymer P1 and P2 Films

The microstructure of P1 and P2 in annealed films was investigated via atomic force microscopy (AFM). The annealed films of P1 and P2 show good crystallinity with no significant phase separation, which facilitates conductive hole migration. Figure 7a shows that the morphology of polymer P1 is a fibrous network structure with a small root mean square roughness (R_q_) of 0.88 nm. The morphology of polymer P2 is not significantly different from that of P1, the only difference being that it has a larger R_q_ value of 1.35 nm (Figure 7b). The ordered arrangement of the crystal domains is relatively more favorable to improvements in the electrical conductivity. The 3D topography of the polymer is shown in Appendix A.

## 3. Discussion

This work introduces weak interaction forces through a molecular structure design strategy, which facilitates the planarity and structural regularity of the material. Suzuki polymerization conditions typically require the introduction of water to dissolve the base and facilitate the reaction, thus limiting the polymerization temperature and efficiency. Both P1 and P2 are fully polymerized at temperatures up to 130 °C and have a high yield, which enables large volumes of such materials to be prepared, e.g., gram scale. Weak force-based strategies are often distinguished from the covalent bond modulation of the molecular plane, which is more prominent for the stretching of materials. The attraction of the non-bonded form offers flexibility, facilitates the recovery of the material after mechanical stress, and offers the possibility of self-healing with resistance to tensile uplift. We believe that other materials designed based on the principle of intramolecular conformational locking could also provide inspiration for the further exploration of device processing and preparation in organic electronics.

## 4. Materials and Methods

Materials: Raw materials such as Pd_2_(dba)_3_ and tri-o-tolylphosphane (P(o-tol)_3_) and organic solvents such as methanol, hexane, etc., were purchased from Sigma-Aldrich (St. Louis, MO, USA). The dried chlorobenzene (water ≤ 50 ppm) was preserved using molecular 3A sieves. 5,7-bis(trimethylstannyl)-2,3-dihydrothieno [3,4-b][1,4]dioxine (EDOT-SnBu_3_) was acquired from SunaTech (Suzhou, China). Both DPP-Br-C8C10 and DPP-Br-C10C12 were prepared from DPP units via alkylation and bromination reactions [49,50]. 

Polymerization method of P1: DPP-Br-C8C10 (210.0 mg, 0.2 mmol), EDOT-Sn (96.3 mg, 0.2 mmol), Pd_2_(dba)_3_ (3.7 mg, 4.0 μmol), and P(o-tol)_3_ (5.0 mg, 16.0 μmol) were added to a Schlenk tube under argon protection. Subsequently, anhydrous chlorobenzene solvent (11.0 mL) was added to the mixture. The reaction mixture was reacted at 130 °C and stirred for 12 h. The reaction was terminated and the mixture was gradually brought to room temperature and dropped into a beaker of methanol (200 mL) using a pipette. The powders are obtained via filtration and subsequently loaded into a Soxhlet extractor. The solvent extraction with methanol, acetone and hexane was carried out sequentially for 9 h to remove the oligomers. The extraction was followed by chloroform and the organic solution was concentrated and precipitated again dropwise in methanol (150 mL). The solids were filtered and then dried under vacuum (90 °C) to afford a black polymer (P1, 88% yield). The preparation of P2 was similar to that of P1, with a yield of 79%.

Characterization: The molecular weight of P1 and P2 was evaluated via gel permeation chromatography (GPC, 150 °C, Agilent PL-GPC220, Atlanta, GA, USA). The eluent used was trichlorobenzene to ensure adequate solubility. The solution at a concentration of 0.1 mg/mL was passed through the column (2× PLgel 10 μm MIXED-B) at a flow rate of 1.0 mL/min. Polystyrene and polystyrene–divinylbenzene were used as the standards and stationary phases, respectively. Tests of the thermal stability of P1 and P2 were conducted under nitrogen (TGA 550, Tainstruments, New Castle, DE, USA). Elemental analysis was conducted using the CHN mode of an organic elemental analyzer (Elementar Vario, Langenselbold, Germany). Photochemical characterization was performed on a UV–visible spectrometer (Cary 5000, Agilent, Santa Clara, CA, USA). The concentration of the polymer solution in the chloroform solvent was around 0.5 mg/mL for testing in the solution state. The solution was further spin-coated onto a clean quartz plate (1.0 cm × 2.0 cm) and the solvent was evaporated via annealing to prepare a thin film for testing in the solid state. Electrochemical tests of the thin films were carried out in acetonitrile solutions containing tetrabutylammonium hexafluorophosphate as the electrolyte. An Ag/AgCl electrode, glassy carbon electrode and platinum electrode were applied as reference electrodes, working electrodes and counter electrodes, respectively. A 1.0 mg/mL polymer chloroform solution was prepared and 6.0 µL of it was pipetted onto a glassy carbon electrode and allowed to evaporate slowly to form a dense film. The HOMO and LUMO energy levels were estimated by extracting data from the potentials at the onset of the oxidation peak and the onset of the reduction peak, respectively, according to the following equations: E_HOMO_ = −4.80 − (E_ox_^onset^ − 0.40) eV; E_LUMO_ = −4.80 − (E_red_^onset^ − 0.40) eV, where E^onset^ is the potential value of the oxidation or reduction measured with respect to the Fc/Fc^+^ redox pair of the measured oxidation or reduction potential value. Thermally annealed films that were prepared under the same conditions as those used for OFET testing were measured with an atomic force microscope to observe their morphology (Nanoscope V, Bruker, Ettlingen, Germany).

Device preparation: Highly doped n-type silicon (Si) wafers with a 300 nm silicon dioxide (SiO_2_) content were used as substrates. The substrates were cleaned via ultrasonic cleaning in deionized water, acetone and isopropanol for 5 min, respectively. Octadecyltrichlorosilane (OTS) was used to modify the substrate. The gold was deposited as source and drain electrodes via photolithography. For the preparation of the semiconductor films, either P1 or P2 was pre-dissolved in chlorobenzene at a concentration of 6 mg/mL and heated overnight at 70 °C with stirring to ensure adequate solubility. The corresponding polymer solutions were individually spin-coated onto the substrates at 2000 rpm (60 s) and then annealed in a glove box under nitrogen at 180 °C for 20 min.

The hole mobility of the polymer-based OFET was characterized using the semiconductor 4200 characterization system (Keithley, USA) operating in the nitrogen glove box. The saturation hole mobility (*µ*) was calculated using the following formula: *μ* = ($\frac{\partial \surd |I_{DS}|}{\partial V_{GS}}$)^2^ ∙ $\frac{2L}{\mathrm{WCi}}$ (I_DS_, V_GS_*,* L and W are the source–drain current, gate voltage, channel length and channel width, respectively; Ci is the dielectric layer capacitance per unit area).

## 5. Conclusions

In this study, two polymers with different side chains were obtained via Stille coupling polymerization. Calculations based on theoretical simulations verified that the coplanar nature of the materials was due to weak intramolecular forces. The side chains affected the weight-average molecular weight of the products and the catalytic efficiency, whose mechanism remains unclear. The effect of the different conjugation lengths on the thermal stability and the energy levels of the frontier orbitals of the materials was insignificant. On the other hand, different side chain lengths led to variations in the stacking of the material from chain to chain, which caused a significant difference in its absorption in the film state. Conductivity tests based on organic transistors showed significantly better hole mobility for polymer P2 compared to P1. Our study explores the effect of side chains on the material properties from a microscopic point of view, which aids in the further design of suitable polymer semiconductor materials. Research work on stretchable organic transistors and sensors for this range of π-conjugated polymers is still ongoing.

## Data Availability

Data is contained within the article.

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
