# Peer review of "Theoretical and Experimental Study of Different Side Chains on 3,4-Ethylenedioxythiophene and Diketopyrrolopyrrole-Derived Polymers: Towards Organic Transistors"

_ijms, 2024, doi:10.3390/ijms25021099_

Round 1
Reviewer 1 Report
Comments and Suggestions for Authors
This manuscript describes the synthesis of two polymeric materials, both of which are based on the most widely researched topic of 3,4-Eth-ylenedioxythiophene (EDOT) in the field of organic electronics. EDOT is polymerized with the electron acceptor unit diketopyrrolopyrrole-thiophene via the Stille coupling and is investigated by varying the length and type of side chain, respectively. Theoretical calculations were excellent and provided a basis for further understanding of the molecular arrangement of such materials. Electrochemical and photophysical tests assisted in comparing the effect of the side chains on the physical properties of the material. Before the manuscript is accepted, I suggest the following revisions.
1. The introduction section should discuss more about the devices that utilize EDOT as core building blocks. Additionally, the research objectives of this manuscript need to be further emphasized.
2. Figure 1 is almost identical to the content of Scheme 1. It is suggested to replace the figure 1 in the introduction section with a figure illustrating the analysis of weak intramolecular forces to visually highlight the key aspects of the study.
3. DSC testing of the polymers needs to be supplemented and the results of IR profiling must be further analyzed.
4. The procedure and details of sample preparation for UV and CV were not mentioned in the main text, which is necessary to clarify it additionally.
5. The DFT section: Atomic coordinates (including imaginary frequency numbers and total energies) and results for HOMO-1 and LUMO+1 need to be supplemented, and the simulated UV measurements need to be supplemented and clarified.
6. I would suggest AFM or XRD testing to further compare the differences in the microscopic characteristics of the materials.
7. The presentation style of the images is inappropriate. Please do not use raw data. Please replot Figure 2c and 2d. In Figure 3, please place the absorption spectra of the same material in one image for comparison.
Comments on the Quality of English Language
Minor grammatical errors need to be corrected.
Reviewer 2 Report
Comments and Suggestions for Authors
The manuscript entitled: "Theoretical and Experimental Based Study of Two Different Side Chains on DPP-EDOT Derived Polymers " is an interesting work on the topic of copolymers based on thiophene, 3,4-Ethylenedioxythiophene (EDOT) and diketopyrrolopyrrole (DPP) obtained by Stille coupling polycondensation and their properties. The results of the synthesis are well presented and the information included in the manuscript regarding the thermal and other characteristics are well documented and based on the experimental and theoretical calculations. I suggest the acceptance of this work after minor modifications to the points:
- Please optimize Figure 2. It is very difficult to identify the values, it is better to remove and to include only the important ones.
- What is the comparison of the synthesis route used up to now compared to the newly proposed mechanism in correlation to the properties of the polymers?
- It should be stated clearly how scalable this methodology is.
Reviewer 3 Report
Comments and Suggestions for Authors
Please, find the comments and suggestions for the authors in the attachment.

The English language is adequate, just minor editing are necessary.
Round 2
Reviewer 3 Report
Comments and Suggestions for Authors
The authors accepted all the reviewers' suggestions increasing substatially the quality of the manuscript, and widening the paper readership. All the authors' answers are convincing and detailed. For these reasons the manuscript should be accepted in present form.
Comments on the Quality of English LanguageThe English language is adequate.